# Equally Good Neurological, Growth, and Health Outcomes up to 6 Years of Age in Moderately Preterm Infants Who Received Exclusive vs. Fortified Breast Milk—A Longitudinal Cohort Study

**DOI:** 10.3390/nu15102318

**Published:** 2023-05-15

**Authors:** Jenny Ericson, Fredrik Ahlsson, Dirk Wackernagel, Emilija Wilson

**Affiliations:** 1School of Education, Health and Social Studies, Dalarna University, 791 88 Falun, Sweden; 2Centre for Clinical Research Dalarna, Uppsala University, 791 82 Falun, Sweden; 3Department of Pediatrics, Falu Hospital, 791 82 Falun, Sweden; 4Department of Women’s and Children’s Health, Uppsala University, 752 36 Uppsala, Sweden; fredrik.ahlsson@kbh.uu.se; 5Division of Neonatology, Department of Pediatrics, University Medical Center of the Johannes Gutenberg, University Mainz, 55131 Mainz, Germany; dirk.wackernagel@unimedizin-mainz.de; 6Department of Clinical Science, Intervention and Technology (CLINTEC), Karolinska Institutet, 171 77 Stockholm, Sweden; 7Department of Women’s and Children’s Health, Karolinska Institutet, 171 77 Stockholm, Sweden; emilija.wilson@ki.se

**Keywords:** development, exclusive breast milk, growth, health, human milk fortification, moderately preterm infant

## Abstract

Moderately preterm infants (32–36 weeks of gestational age) have an increased risk of worse health and developmental outcomes compared to infants born at term. Optimal nutrition may alter this risk. The aim of this study was to investigate the neurological, growth, and health outcomes up to six years of age in children born moderately preterm who receive either exclusive or fortified breast milk and/or formula in the neonatal unit. In this longitudinal cohort study, data were collected for 142 children. Data were collected up to six years of age via several questionnaires containing questions about demographics, growth, child health status, health care visits, and the Five to Fifteen Questionnaire. Data on the intake of breast milk, human milk fortification, formula, and growth during hospitalization were collected from the children’s medical records. No statistically significant differences in neurological outcomes, growth, or health at six years of age were found between the two groups (exclusive breast milk, n = 43 vs. fortified breast milk and/or formula, n = 99). There is a need for research in larger populations to further assess potential effects on health and developmental outcomes when comparing the use of exclusive versus fortified breast milk for moderately preterm infants during neonatal hospitalization.

## 1. Introduction

Moderately preterm infants (32–36 weeks of gestational age) are a vulnerable group for which research has shown worse neurological development and an increased risk of the development of metabolic health problems later in life compared to infants born at term [1,2]. From a clinical and research perspective, this group is disregarded even though it is the largest group of preterm infants. Although moderately preterm infants have a lower incidence of morbidity than very preterm infants, the larger population will lead to a greater number of children with health-related problems.

Nutrition is an important factor in facilitating child growth, health, and development. Breast milk is the recommended source of nutrition for preterm infants [2]. Breast milk has been shown to decrease the risk of numerous infections, diseases and mortality compared to infant formula. Breast milk facilitates the immune system and neurological and metabolic development in children [3,4]. Furthermore, breastfed infants also have improved cognitive development and an increased IQ compared to formula-fed infants [1,3,4,5]. There is a dose-response effect with more benefits from exclusive breastfeeding and breastfeeding over a longer duration [6,7,8,9].

Pediatric societies recommend that preterm infants should experience intrauterine-like growth postnatally [10]. However, there is an ongoing discussion as to whether this is a reasonable recommendation [10,11]. Growth charts for preterm infants are based on cross-sectional birth weight data, which do not reflect normal postnatal growth for preterm infants. An earlier study revealed that healthy preterm infants showed an average weight z score of −0.8 after postnatal weight loss [11]. Compared to infants born at term, preterm infants have an increased risk of developing energy, protein, mineral, vitamin, and trace element deficiencies during hospitalization, which may lead to suboptimal postnatal growth [12,13]. Suboptimal postnatal growth is associated with delayed neurological and cognitive development [14,15]. Sometimes, exclusive breast milk does not fulfill the recommended nutrition intake. One often used strategy is to fortify breast milk with extra nutrients in an attempt to facilitate infant development and growth. Potential risks with fortification are increased density, osmolality, and oxidative stress, which may result in feeding difficulties and increased risk of sepsis, inflammation, and worse neurological outcomes in preterm infants [16,17,18]. Furthermore, accelerated growth during early life may be associated with altered body fat distribution, which has been related to metabolic consequences such as an increased risk of insulin resistance and high blood pressure [19,20,21].

Currently, there is no evidence available about the effect of human milk fortification on moderately preterm infants. However, several studies have shown that infants receiving breast milk during hospitalization and at discharge have a better neurological outcome at two and five years of age compared to infants receiving formula [22,23,24], despite a lower weight gain during hospitalization [25]. The lack of scientific evidence on human milk fortification may lead to both over- and undernutrition and thereby a risk of unhealthy development later in life. The aim of this study was to investigate the neurological, growth, and health outcomes up to six years of age in children born moderately preterm who receive exclusively breast milk in the neonatal unit compared to children who receive fortified breast milk and/or formula.

## 2. Materials and Methods

### 2.1. Setting

Participants were recruited from six neonatal units geographically spread across Sweden. All neonatal units started infant enteral feeding as soon as possible after birth, often within the first few hours. The aim was a fluid volume of 150–180 mL/day after approximately five days, in some cases up to 200 mL/day. During the study period of 2013–2015, three participating neonatal units did not routinely fortify breast milk for moderately preterm infants. The remaining three neonatal units routinely fortified breast milk, either for infants born at gestational week 32 or for all infants born before gestational week 35 (Table 1 and Figure 1). These units performed nutritional analyses on each mother’s breast milk and on all donor milk and used the calculating program Nutrium [26] to provide targeted fortification for all infants. All neonatal units had their own breast milk bank or access to a central breast milk bank. Depending on access to donor breast milk or traditions in the unit, the infants born near term received preterm formula or formula instead of donor milk during the first days or when their own mother’s milk was lacking.

### 2.2. Population

Parents of children born moderately preterm (32–36 weeks of gestational age) were asked to participate in this study by a letter sent to the families’ homes (n = 180) when their children were six years old, and 142 (78%) consented to participate in the study. The families had previously participated in a randomized controlled trial (RCT) [27] in which they had agreed to be contacted for future research projects. The inclusion criterion for the RCT was that the children received some breast milk at discharge from the neonatal unit.

### 2.3. Data Collection

A questionnaire containing questions about child health status, parental weight and height, child growth, health care visits, and the Five to Fifteen Questionnaire [28] was sent to the participating parents when their child was six years old. Data about daily intake of breast milk, human milk fortification, formula, glucose infusions, partial parenteral nutrition, growth, and first day of free feeding during hospitalization were collected from the children’s medical records. Demographic data and data about the children’s health, growth, breastfeeding status, and health care visits after discharge from the neonatal unit up to one year of age were collected via questionnaires sent to the families during the original RCT study [27]. For detailed information on data collection and the recruitment of participants, see Figure 1.

The Five to Fifteen Questionnaire is a validated Nordic questionnaire that targets developmental and behavioral problems and consists of 152 questions regarding children aged six years (the learning domain is excluded). The answering options are “does not apply” (0), “applies sometimes or to some extent” (1), or “definitely applies” (2). For children aged six years, the items are arranged into 7 general domains: motor skills, executive functioning, perception, memory, language, social skills, and emotional/behavioral problems. The domains can be further divided into 16 subdomains investigating gross and fine motor skills, attention, hyperactivity, hypoactivity, planning, and perception of space, time, and body, as well as visual perception, comprehension, expressive language skills, communication, internalizing and externalizing behaviors, and obsessive-compulsive behavior. The Five to Fifteen Questionnaire domains and subdomains have been shown to have acceptable to high internal consistency and good test-retest reliability [28]. A value above the 90th percentile indicates definite problems. The cut-off originates from a normative sample consisting of Swedish children aged 6 to 15 years [28].

All weight, length, and head circumference measurements from birth to approximately 5.5 years of age were collected from the children’s medical records and the questionnaires.

All questionnaires included questions about sickness (yes/no) and general health or wellbeing. The questionnaires also included questions about health care visits in between the study follow-ups. The questionnaire completed at six years of age also included a question about gastrointestinal problems (yes/no).

### 2.4. Statistical Analysis

The participants were divided into two groups based on the feeding protocol during the hospital stay: the exclusive breast milk group and the fortified breast milk and/or formula group.

Negative binomial models were used to compare Five to Fifteen Questionnaire domain and subdomain scores between the two feeding groups. The models are presented with coefficients, 95% confidence intervals (95% CIs), and *p* values. In addition, the odds ratios using binary outcomes of definite developmental problems (no problems vs. problems) were compared between the two feeding groups using logistic regression models for each domain. The models are also presented with 95% confidence intervals (95% CIs), and *p* values. Adjustments were made for gestational age at birth, child sex, and maternal educational level (higher education vs. high school and lower) in all regression models.

Growth data were transformed to standard deviation scores (SDSs) for weight, length, and head circumference using Swedish norm data [29]. The association between the two feeding groups and child growth was analyzed in linear mixed-effects models. The models included fixed effects for the children’s age, feeding group, interactions, and parental anthropometry. Furthermore, the models also included a random intercept for each child and an autoregressive correlation structure for the repeated measures. Child age was included with restricted cubic splines to allow a nonlinear association with the outcome in the weight model. Based on this model, the marginal means, 95% CIs, and *p* values are presented per time point and feeding group. The time points were set at 36 and 40 weeks of gestational age, 6 months of age, and 1, 1.5, 2.5, 4, and 5.5 years of age. Adjustments were made for gestational age at birth, child sex and parental anthropometry in all models.

The questions about child sickness and gastrointestinal problems were analyzed with descriptive statistics. To investigate if there were possible differences between the two feeding groups, a Chi2 test was performed for the respective question and the *p* value is presented. The children’s general health and wellbeing were estimated by their parents on a Likert scale ranging from 1 (“bad”) to 5 (“excellent”). The data were descriptively analyzed and presented as the mean and standard deviation (SD). An independent t test was performed for each follow-up time point to calculate possible differences between the two feeding groups, and *p* values are presented. Health care visits were divided into primary health care visits (i.e., visits to general practitioners/outpatient centers), hospital admissions and emergency unit visits. The number of visits to each health care level up to six years of age was analyzed, and the results are presented as numbers, percentages, means and SDs. Differences were analyzed using an independent t test for each health care level, and *p* values are presented. To calculate possible differences between the two feeding groups in the number of children who had a health care visit, a Chi2 test was performed, and *p* values are presented. Data were analyzed with IBM SPSS Statistics for Windows (version 28.0, IBM Corp., Armonk, NY, USA) and R (R Core Team, 2022. R Foundation for Statistical Computing, Vienna, Austria).

## 3. Results

In total, 142 children participated in the study; the demographics of the participants are presented in Table 2. The distribution of the participating children from the six different neonatal units can be obtained from Table 1.

### 3.1. Neurodevelopmental Outcomes

There were no differences between the two feeding groups in the proportion of children defined as having definite problems or not (Table 3), in the cumulative scores (Table 4) or in any of the Five to Fifteen Questionnaire domain or subdomain scores.

### 3.2. Growth

The children in both feeding groups in this study had weight and height SDSs that were slightly lower than the Swedish mean SDSs up to six years of age. The head circumference SDSs were not different from the Swedish mean SDSs in the breast milk group and slightly above the Swedish mean SDSs in the fortified breast milk and/or formula group (Appendix A and Figure 2). No statistically significant differences were found between the two feeding groups in weight, height, or head circumference SDSs (Appendix A and Figure 3).

### 3.3. Health

The children were reported to have varying degrees of sickness during their first six years of life. No statistically significant differences were observed between the two feeding groups (Table 5). Regarding gastrointestinal problems, 21 (49%) children in the exclusive breast milk group and 43 (43%) children in the fortified breast milk and/or formula group had gastrointestinal problems, and there was no statistically significant difference between the two feeding groups (*p* = 0.34, results not shown). No statistically significant differences were found between the two feeding groups concerning the children’s general health or wellbeing at any of the follow-up time points (Table 5). The number and percentage of children who had health care visits at the three different health care levels are presented in Table 6 together with the mean number of visits at each health care level. No statistically significant differences were found between the two feeding groups (Table 6).

## 4. Discussion

We did not observe any statistically significant differences in neurological, growth, or health outcomes up to six years of age in children born moderately preterm who received either exclusive breast milk or fortified breast milk and/or formula during hospitalization in the neonatal unit.

No significant differences were found in any of the Five to Fifteen Questionnaire dimensions or subdimensions between the two feeding groups regarding the proportion of children who had problems or not or in the difference in total scores. It is uncertain how much impact a short period of exposure (a few days to three weeks) had on the outcomes. It seems probable that human milk fortification was not superior to breast milk alone for these children. Likewise, many of the children who received formula in addition to breast milk only received formula for a few days. Thus, their exposure period was also short. However, regarding children who were partially breastfed at discharge, the results from the original RCT showed that these children had an increased risk of ceasing breastfeeding earlier than children who were exclusively breastfed at discharge [30]. This should be considered when choosing supplementary feeding for infants in the neonatal unit.

Human milk fortification did not increase growth in this study. This finding differed from those of a Cochrane review showing that human milk fortification for preterm infants led to a small increase in weight gain, length, and head circumference. However, the follow-up periods were short, and most included studies investigated very preterm infants [31]. Our results showed that children in both groups weighed slightly below the Swedish mean SDSs up to the age of six years. An earlier study [11] revealed that preterm infants showed an average weight z score of −0.8 after postnatal weight loss and that it is common and expected for some healthy preterm infants to have a weight z score below 0 at hospital discharge, which the children in our study also showed. The children’s length was approximately half an SDS below the Swedish mean SDS, and the head circumference was almost similar to the Swedish mean, which indicated good growth in both feeding groups. The children in the study had quite a rapid start of enteral feeding. In addition, the volumes increased fast, and the target volumes were high, which may have contributed to our result on child growth.

Our study did not reveal any differences between the two feeding groups in sickness, gastrointestinal problems, general health and wellbeing or health care visits. The infants included in this study were quite healthy during hospitalization and appeared to continue to be healthy, at least at the group level.

### Limitations

A major limitation in this study was the observational design, which limited the interpretation of causality between exposures and outcomes. In addition, the sample size was relatively small. The children in the study were quite healthy, had no neonatal illnesses or major malformations, and had highly educated mothers, which should be considered when interpreting the results. The Five to Fifteen Questionnaire was developed to identify children with developmental and behavioral problems and may not be the best scale for measuring neurological outcomes. However, due to practical reasons of feasibility, the scale had to be parent-reported, have a Swedish translation, and be intended for children at six years of age. Thus, the Five to Fifteen Questionnaire suited our needs. Furthermore, we chose to use Swedish norm growth data, despite the reported disadvantages in plotting errors and the exclusion of many infants [32]. The advantages of using the same growth norm on the follow-up occasions outweighed the disadvantages. However, the results need to be interpreted with this in mind. In this study, data were lacking on the children’s metabolic health, which is also important to investigate because earlier studies have shown metabolic consequences for children receiving high amounts of protein early in life [33].

## 5. Conclusions

The use of fortified breast milk and/or formula during hospitalization in the neonatal unit had no statistically significant association with moderately preterm-born children’s neurological, growth, or health outcomes up to six years of age compared to the use of exclusive breast milk. More research is needed in larger populations of moderately preterm infants to assess the potential effects, i.e., the benefits versus risks, of providing exclusive breast milk versus fortified breast milk during neonatal hospitalization.

## Figures and Tables

**Figure 1 nutrients-15-02318-f001:**
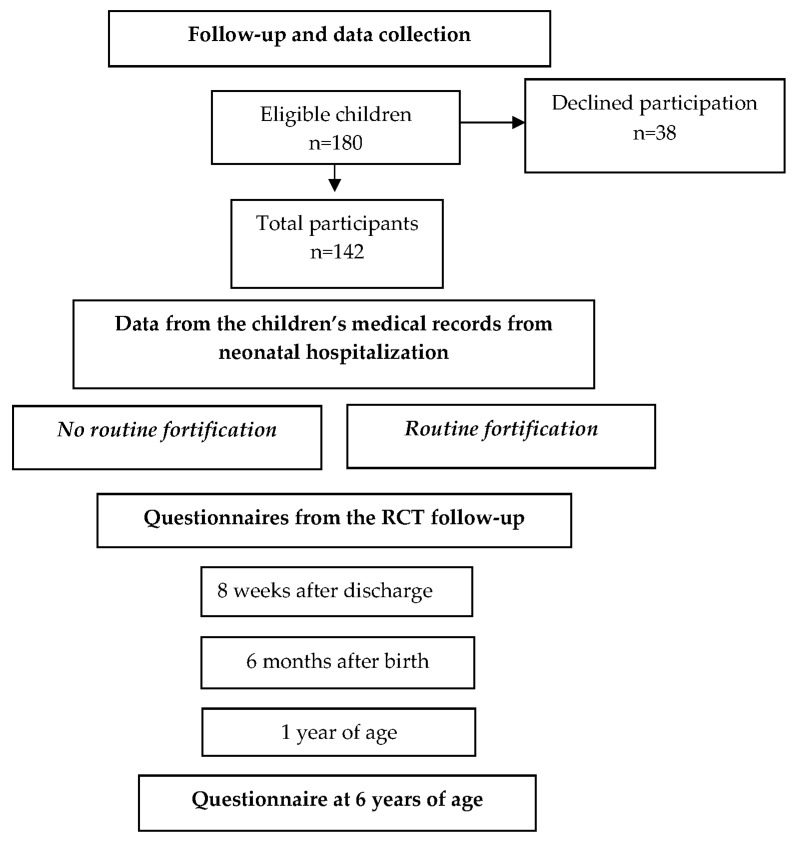
Follow-up and data collection in the study.

**Figure 2 nutrients-15-02318-f002:**
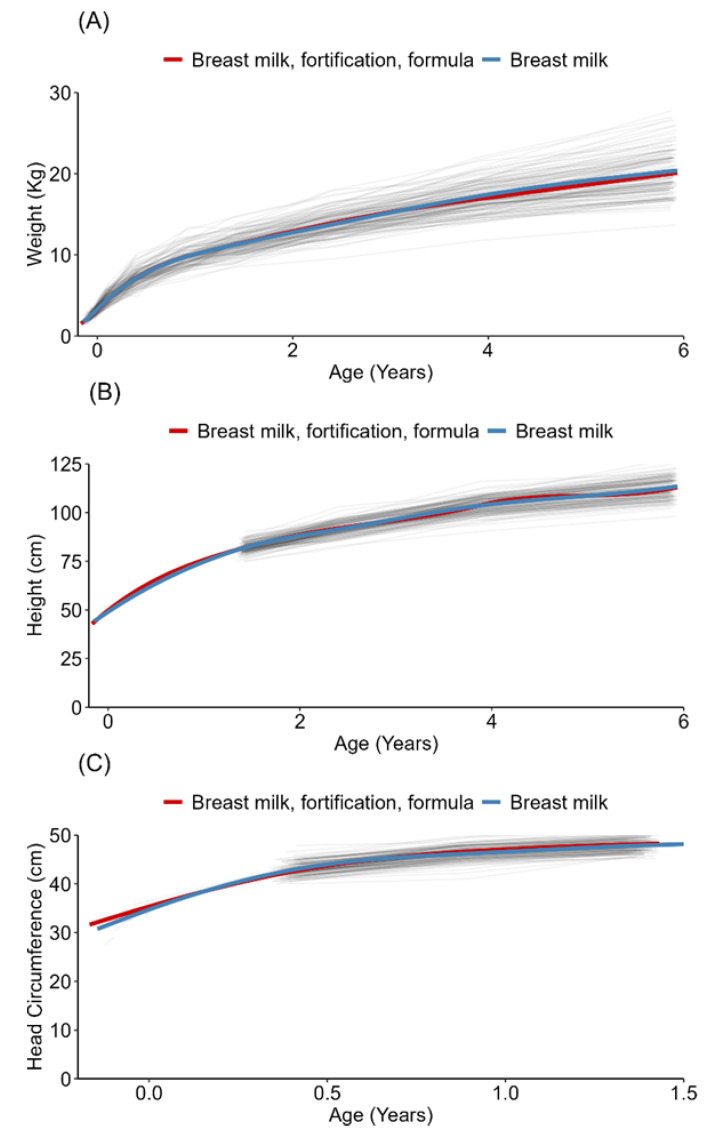
Growth in the two feeding groups from birth up to 6 years of age (**A**) weight gain (grams); (**B**) height (centimeters); (**C**) head circumference (centimeters) up to 1.5 years of age.

**Figure 3 nutrients-15-02318-f003:**
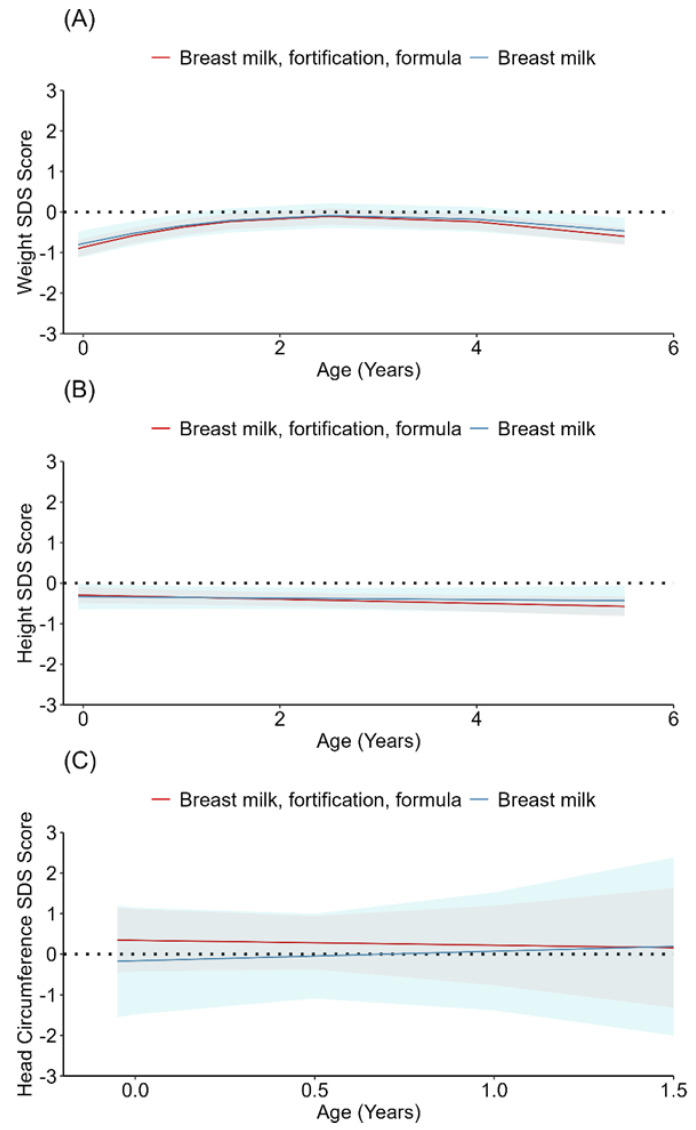
SDSs in the two feeding groups from birth up to 6 years of age (**A**) weight; (**B**) height; (**C**) head circumference up to 1.5 years of age.

**Table 1 nutrients-15-02318-t001:** The number and percentage of participating children from each neonatal unit and in the respective feeding groups.

	Total	Breast Milk (n = 43)	Fortified Breast Milk and/or Formula (n = 99)	Routine Fortification
	n (%)	n (%)	n (%)	
Neonatal unit 1	35 (25)	10 (29)	25 (71)	Yes
Neonatal unit 2	20 (14)	0	20 (100)	Yes
Neonatal unit 3	26 (18)	11 (42)	15 (58)	No
Neonatal unit 4	31 (22)	14 (45)	17 (55)	No
Neonatal unit 5	19 (13)	4 (21)	15 (79)	Yes
Neonatal unit 6	11 (8)	4 (36)	7 (64)	No

**Table 2 nutrients-15-02318-t002:** Demographics of the participating children (n = 142) and their mothers (n = 131) in the total group and in each feeding group. Presented with numbers and percentages (%), mean and standard deviation ±SD, median and intra quartile range [IQR] and *p* value.

	Total	Breast Milk (n = 43)	Fortified Breast Milk and/or Formula (n = 99)	*p* Value
Maternal age at infant birth (years)	31.1 ± 4.6	29.8 ± 4.5	31.7 ± 4.5	0.02
Higher education	103 (73)	27 (63)	76 (77)	0.07
Primipara	72 (51)	26 (63)	46 (47)	0.05
Mother born in another country	10 (7)	2 (5)	8 (8)	0.37
Cesarean section	60 (42)	16 (37)	44 (44)	0.27
Gestational age at birth (weeks)	34 [2]	34 [1]	34 [2]	0.84
Multiple births	11 (8)	0	11 (12)	0.01
Male	74 (52)	19 (44)	55 (56)	0.14
Small for gestational age	8 (6)	2 (5)	6 (6)	0.54
Birth weight (grams)	2445 ± 466	2474 ± 428	2438 ± 483	0.68
Birth length (centimeters)	45.9 ± 2.8	46 ± 2.6	45.9 ± 2.6	0.87
Birth head circumference (centimeters)	32.3 ± 1.6	32.3 ± 1.4	32.2 ± 1.7	0.68
Neonatal illness (NEC, ROP, PVL, BPD, congenital malformation)	0	0	0	
Antibiotics	19 (13)	7 (16)	12 (12)	0.34
Hypoglycemia	49 (36)	15 (37)	34 (36)	0.56
Glucose infusion	42 (30)	14 (33)	28 (29)	0.37
Partial parental nutrition	9 (6)	4 (9)	5 (5)	0.28
Donated breast milk	117 (82)	42 (98)	75 (76)	<0.001
Fortification	33 (23)	0	33 (33)	
Number of days with fortification				
Formula	74 (52)	0	74 (75)	
Exclusive breast milk	43 (30)	43 (100)	0	
First day of free feeding	17 [12.25]	15 [8]	18 [13]	0.03
Hospital stay (days)	22 [13]	20 [11]	22 [13]	0.09
Exclusive breastfeeding at discharge	127 (89)	42 (98)	85 (86)	0.03
Exclusive breastfeeding 8 weeks after discharge	102 (72)	35 (81)	67 (68)	0.07
Exclusive breastfeeding 6 months after birth	41 (29)	13 (30)	28 (28)	0.57
Breastfeeding at one year of age	18 (13)	3 (7)	15 (15)	0.14

**Table 3 nutrients-15-02318-t003:** Number and percentage of children with definite problems (>90 percentile) and no problems in each of the seven domains of the Five to Fifteen Questionnaire and the two feeding groups. Thereto, a logistic regression model of problems vs. no problems and feeding group in the respective domain. Odds ratios (ORs), 95% confidence intervals (95% CIs) and *p* values are presented.

	Total		Breast Milk (n = 43)		Fortified Breast Milk and/or Formula (n = 99)		Logistic Regression Model *	
	No Problem	Problems	No Problem	Problems	No Problem	Problems	OR 95% CI	*p* Value
Motor skills	126 (89)	15 (11)	39 (91)	4 (9)	88 (89)	11 (11)	1.20 (0.35–4.15)	0.77
Executive functioning	135 (95)	7 (5)	40 (93)	3 (7)	95 (96)	4 (4)	0.64 (0.13–3.23)	0.59
Perception	124 (87)	18 (13)	39 (93)	4 (9)	85 (86)	14 (14)	1.89 (0.57–6.34)	0.30
Memory	133 (94)	9 (6)	41 (95)	2 (5)	92 (93)	7 (7)	1.48 (0.29–7.64)	0.64
Language	122 (86)	20 (14)	38 (88)	5 (12)	84 (4)	15 (15)	1.66 (0.52–5.31)	0.39
Social skills	134 (94)	8 (6)	42 (98)	1 (2)	92 (93)	7 (7)	2.81 (0.32–24.6)	0.35
Emotional/behavioral problems	129 (91)	13 (9)	41 (95)	2 (5)	88 (88)	11 (11)	2.46 (0.48–12.5)	0.46

* Adjusted for gestational age at birth, child sex and maternal educational level.

**Table 4 nutrients-15-02318-t004:** Negative binominal regression model of the Five to Fifteen Questionnaire cumulative scores in the respective domain and feeding group. Estimated negative binomial regression coefficients, 95% confidence intervals (95% CIs) and *p* values are presented. Adjustments were made for gestational age at birth, child sex and maternal educational level.

	Breast Milk (n = 43)	Fortified Breast Milk and/or Formula (n = 99)	
		Coefficients (95% CI)	*p* Value
Domain			
Motor skills	ref	1.25 (0.78–1.98)	0.35
Executive functioning		1.08 (0.71–1.65)	0.71
Perception		1.17 (0.76–1.80)	0.49
Memory		1.41 (0.85–2.35)	0.18
Language		1.49 (0.93–2.39)	0.10
Social skills		1.10 (0.69–1.78)	0.69
Emotional/behavioral problems		0.92 (0.59–1.44)	0.72
Subdomain			
Gross motor skills		1.32 (0.78–2.30)	0.33
Fine motor skills		1.19 (0.73–1.95)	0.48
Attention		1.11 (0.70–1.76)	0.65
Hyperactivity		0.98 (0.63–1.54)	0.93
Hypoactivity		0.79 (0.42–1.49)	0.46
Planning		1.64(0.86–3.02)	0.13
Perception of space		1.12 (0.58–2.13)	0.74
Perception of time	ref	1.08 (0.69–1.70)	0.74
Perception of body		1.62 (0.85–3.09)	0.14
Visual body perception		0.89 (0.39–2.05)	0.78
Comprehension		1.33 (0.73–2.43)	0.35
Expressive speech		1.46 (0.86–2.48)	0.17
Communication skills		1.42 (0.70–2.88)	0.33
Internalizing behavior		0.75 (0.44–1.27)	0.28
Externalizing behavior		0.97 (0.59–1.58)	0.89
Obsessive compulsive behavior		1.74 (0.79–3.82)	0.17

**Table 5 nutrients-15-02318-t005:** Sickness and general health and wellbeing at 8 weeks after discharge from the neonatal unit and 6- and 12-months postnatal age and general health at 6 years of age as reported by parents.

	Sickness				General Health and Wellbeing			
	Total	Breast Milk (n = 43)	Fortified Breast Milk and/or Formula (n = 99)		Total	Breast Milk (n = 43)	Fortified Breast Milk and/or Formula (n = 99)	
	n (%)			*p* Value	Mean ± SD			*p* Value
Eight weeks after discharge (n = 129)	51 (40)	15 (36)	36 (41)	0.34	4.3 ± 0.69	4.3 ± 0.63	4.3 ± 0.73	0.66
Six months after birth (n = 124)	71 (57)	21 (54)	50 (59)	0.37	4.5 ± 0.64	4.7 ± 0.58	4.5 ± 0.67	0.14
One year of age (n = 83)	70 (84)	22 (79)	48 (87)	0.24	4.6 ± 0.51	4.8 ± 0.44	4.6 ± 0.53	0.08
Six years of age (n = 140)	95 (68)	29 (67)	66 (68)	0.55	4.6 ± 0.68	4.5 ± 0.70	4.6 ± 0.67	0.81

**Table 6 nutrients-15-02318-t006:** The number and percentages of children that had a primary health care visit (in addition to the usual child health care follow-up program), hospital care visit or an emergency visit up to 6 years of age, presented in total and for each feeding group. The mean, standard deviation (SD) and a *p* value of the number of visits among the children that had a health care visit in total and for each feeding group.

	Total		Breast Milk (n = 43)		Fortified Breast Milk and/or Formula (n = 99)			
	Number (%)	Mean ± SD	Number (%)	Mean ± SD	Number (%)	Mean ± SD	*p* Value for Number of Visits	*p* Value Visit
Primary health care	101 (72)	4.67 ± 6.51	28 (65)	3.39 ± 2.19	73 (75)	5.15 ± 7.49	0.27	0.18
Hospital care visit	51 (36)	2.37 ± 2.24	19 (44)	1.85 ± 0.80	32 (33)	2.68 ± 2.73	0.19	0.13
Emergency visit	88 (62)	2.60 ± 2.56	27 (63)	2.10 ± 1.64	61 (62)	2.79 ± 3.00	0.33	0.55

## Data Availability

The datasets generated and analyzed during the current study are not publicly available for ethical and legal reasons but are available from the corresponding author on reasonable request.

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
