# Peer review of "Equally Good Neurological, Growth, and Health Outcomes up to 6 Years of Age in Moderately Preterm Infants Who Received Exclusive vs. Fortified Breast Milk—A Longitudinal Cohort Study"

_nutrients, 2023, doi:10.3390/nu15102318_

Round 1
Reviewer 1 Report
The topic of this work is relevant and attracts interest of the specialists of various fields including nutritional specialists. Moreover, the design of the study was fine (except for the following) and carried out appropriately. In this respect, this manuscript is suitable to be published in Nutrients only if the following correction is appropriately done.
It is highly probable that the size of adverse health outcomes of moderately preterm infants would be small so that the sample size must be large enough to detect statistical difference derived from any intervention. In this respect, the result of this work and the authors’ conclusion could have been predicted from the beginning. However, the authors mention that the sample size is not small in this field (line 339-340): it is not acceptable comment. Even if the statement is true (sample sizes of other similar studies were smaller than this work), it does never mean that small sample size is acceptable.
The authors are requested to delete this description. Instead, they are required to statistically estimate how much sample size is required to obtain statistical significance based on this study results.
Author Response
Dear editor and reviewer
Thank you for your effort to review our manuscript and for your valuable comments. Revisions are marked in yellow in the revised manuscript. Withdrawn texts were not retained as crossed-out text. Please find below our replies to the comments.
Thank you for your comments and feedback. We have revised the sentence. However, we have not performed a post-hoc power calculation thus is not recommended and adds no meaningful information in addition to what is already available by the 95% CI. We refer to Hoening J and Heisy D The Abuse of Power: The Pervasive Fallacy of Power Calculations for Data Analysis 2001, A Note on Misconceptions Concerning Prospective and Retrospective Power on JSTOR, https://stat.uiowa.edu/sites/stat.uiowa.edu/files/techrep/tr378.pdf
Reviewer 2 Report
The content of the report is that both exclusive breast milk and fortified breast milk showed neurologically unproblematic development. Although the results seem reasonable, some modifications are necessary.
1. As a premise, are there neurological differences between children who were born preterm and those who were born full-term?
2. Do Table 5 and Figure 2/3 show the same content? If they are the same, then table 5 should be supplementary or unnecessary. Since there is no significant difference in the data, there seems to be no need to show them.
3. As shown in Table 1, Figure 1, it is difficult to understand the significance of dividing the data into units 1 through 6. Please clarify the difference. If possible, it would be better to describe the difference between each unit in more detail in table 1.
4. As for table 2, there are many missing units. In particular, please specify the units for age, weight, and length. Please explain the meaning of the numbers, () and [] in the table (whether they are numbers or %).
5. Page 7, “Tabell 3” should be corrected.
Grammatical errors should be rechecked.
Author Response
Dear editor and reviewer
Thank you for your effort to review our manuscript and for your valuable comments. Revisions are marked in yellow in the revised manuscript. Withdrawn texts were not retained as crossed-out text. Please find below our replies to the comments.
- As a premise, are there neurological differences between children who were born preterm and those who were born full-term?
Reply: Yes, children born preterm have a higher risk for worse neurological development compared to children born full term. We have written about that in the first paragraph in the introduction.
- Do Table 5 and Figure 2/3 show the same content? If they are the same, then table 5 should be supplementary or unnecessary. Since there is no significant difference in the data, there seems to be no need to show them.
Reply: We have chosen to have figure 2 and 3 in the manuscript and to move table 5 to supplementary material.
- As shown in Table 1, Figure 1, it is difficult to understand the significance of dividing the data into units 1 through 6. Please clarify the difference. If possible, it would be better to describe the difference between each unit in more detail in table 1.
Reply: Thank you for the comment, we have revised table 1 and figure 1.
- As for table 2, there are many missing units. In particular, please specify the units for age, weight, and length. Please explain the meaning of the numbers, () and [] in the table (whether they are numbers or %).
Reply: Thank you for the comment, we have clarified the table.
- Page 7, “Tabell 3” should be corrected.
Reply: Thank you for noticing that. We have corrected the spelling.